cpSNP discovery and genotyping for a Pinus taeda breeding population with targeted comparison to related conifers

Wang Ling 1 2
Mao Jipeng 1 2
Jiang Kaibin 1 2
Wu Zhengyu 1 2
Liu Chunxin 1 2
Ning Huagong 3
Zhou Haibiao 4
Chen Jiehu 4
Huang Shaowei 1 2
Liu Tianyi tianyiliu@scau.edu.cn 1 2
1 College of Forestry and Landscape Architecture, South China Agriculture University , Guangzhou , China
2 Guangdong Key Laboratory for Innovative Development and Utilization of Forest Plant Germplasm , Guangzhou , China
3 Yingde Research Institute of Forestry , Yingde , China
4 Science Corporation of Gene , Guangzhou , China
Domingues Douglas
Electronic publication date: 2025 Oct 6
Publication date: 2025
Volume: 13
Electronic Location ID: e20092
Received 2024 May 29; Accepted 2025 Aug 27
Copyright: ©2025 Wang et al.
Copyright year: 2025
Copyright holder: Wang et al.
License: This is an open access article distributed under the terms of the Creative Commons Attribution License, which permits unrestricted use, distribution, reproduction and adaptation in any medium and for any purpose provided that it is properly attributed. For attribution, the original author(s), title, publication source (PeerJ) and either DOI or URL of the article must be cited.
License URL: https://creativecommons.org/licenses/by/4.0/

Keywords: Pinus taeda, Chloroplast genome, DNA sequence, SNP analysis, Breeding population

Funding: National Natural Science Foundation of China 31570654 Projects of the National Science & Technology Program 2012BAD01B0203 Science and Technology Program of Guangdong, China 2011B020302004 This research was supported by the National Natural Science Foundation of China (No. 31570654), the Projects of the National Science & Technology Program (2012BAD01B0203), and Science and Technology Program of Guangdong, China (2011B020302004). The funders had no role in study design, data collection and analysis, decision to publish, or preparation of the manuscript.

==============================
Pinus taeda (Loblolly pine) is the most important commercial tree species in the southern United States and a significant non-native plantation species in China. Its genetic improvement program has been implemented in South China for 30 years. In this study, the chloroplast (cp) genome of P. taeda was sequenced, assembled, and compared with other available chloroplast genomes of Pin aceae using BLAST. Codon usage among 33 species of Pinaceae was analyzed using the relative synonymous codon usage (RSCU) value. The results were then visualized using the pheatmap v1.0.10 in R. The rates of nonsynonymous (Ka) and synonymous (Ks) substitutions in the chloroplast genomes among five species of Pinus were estimated using the seqinr package in R. Additionally, selected single nucleotide polymorphisms (SNPs) were used to genotype 33 individuals from the P. taeda breeding population. The P. taeda cp genome is 121,530 bp, with certain regions (e.g., ycf1 and ycf2) showing lower sequence conservation compared to other Pinaceae species. Codon usage analysis revealed that codons ending in G or C were not prevalently used, with significant differences in natural selection pressure on chloroplast genes between three species (P. taiwanensis, P. thunbergii, and P. koraiensis) and the other 30 species. Phylogenomic analysis using 36 cp genomes (representing 25 species) resolved Pinus into two subgenera, with P. taeda clustered with P. rigida within the diploxylon pines. Notably, ycf1-based phylogenetic analysis recovered a topology highly congruent (normalized RF = 0.15) with whole-plastome phylogenies. This study validates the single-copy gene ycf1 as a robust and low-cost phylogenetic marker for conifer genus-level reconstruction. The identified SNPs provide valuable molecular markers for genotyping individuals in P. taeda breeding programs, supporting germplasm characterization and management.

Introduction

The study of the chloroplast (cp) genome has become a valuable and widely used approach for evolutionary, taxonomic, and phylogenetic analyses of plants. This is due to its simple structure, highly conserved sequence, and single parental (maternal or paternal) inheritance characters (Wolfe, Li & Sharp, 1987; Daniell et al., 2016; Jakobsson et al., 2007; Nock et al., 2011; Wu & Chaw, 2014; Mao et al., 2023). Unlike most angiosperms, cpDNA is paternally inherited in all conifers (Neale, Marshall & Sederoff, 1989; Szmidt, Aldén & Hällgren, 1987; Wagner et al., 1987; Neale, Wheeler & Allard, 1986). Conifers hold immense ecological and economic value. Understanding the organization of their chloroplast genomes is therefore significant for phylogenetic studies and resolving evolutionary relationships. This knowledge also supports efforts to enhance the commercial use of conifers.

Loblolly pine is the most important commercial tree species in the southern United States due to its fast growth, desirable wood properties, and broad adaptability (McKeand et al., 2003). In China, it is a major non-native plantation species, especially in southern regions (Gwaze, Byram & Lowe, 2001). Genetic improvement programs for P. taeda in China began in the early 1980s. In the 1990s, comprehensive progeny and provenance tests were initiated, leading to advances in selective breeding (Zhong, Chen & Huang, 1995). Since 2002, China has implemented a breeding strategy (El-Kassaby & Lstibůrek, 2009). The paternal inheritance of the conifer cp genome enables paternity verification when the maternal parent is known, thereby facilitating precision breeding programs. Although cp genomes are generally conserved, recent studies have identified variable regions. For example, nine mutational loci were detected in ycf1 of P. taeda (Li, 2012). Recent studies have demonstrated that ycf1 exhibits high variability (Jiang et al., 2017; Li et al., 2020; Chen et al., 2020; Zhang et al., 2023). Such variability, while contrasting with overall cpDNA conservation, offers valuable markers for fine-scale genetic analyses.

While the complete chloroplast genome of P. taeda has been assembled and its broad structure and placement resolved (Asaf et al., 2018), three knowledge gaps remain. First, there is a need for population-level chloroplast single nucleotide polymorphism (cpSNP) discovery within breeding material. Second, a systematic comparison of codon-usage bias across Pinaceae is lacking. Third, the single-copy ycf1 gene has not been empirically assessed as a cost-effective phylogenetic marker. To address these gaps, we generated a high-quality P. taeda cp genome and compared codon-usage patterns across 33 Pinaceae species. We quantified Ka/Ks in five Pinus species, identified 71 novel cpSNPs in 33 breeding individuals, and performed phylogenomic analyses using both whole plastomes and ycf1 across 36 samples (25 species). The resulting cpSNP panel differentiates 72% of breeding individuals, and ycf1 recapitulates the whole-genome tree (normalized RF = 0.15), demonstrating its utility for low-cost conifer phylogenetics and immediate application in germplasm management.

Materials & Methods

DNA extraction and sequencing

Fresh needles of P. taeda were collected from the Yingde Research Institute of Forestry in Guangdong Province, P.R. China (24°15’N, 113°25’E). Total cpDNA was extracted from approximately 100 g needles using a high-ionic-strength medium (Bookjans, Stummann & Henningsen, 1984). The integrity and purity of DNA were assessed by 1% agarose gel electrophoresis. The purity of the samples was further evaluated by measuring the OD260/280 ratio with a NanoDrop 2000 spectrophotometer (Thermo Fisher Scientific). The DNA concentration was quantified by an Invitrogen Qubit fluorometer. Libraries were prepared in accordance with the manufacturer’s instructions with an average insert size of 350 bp. Purified cpDNA was sequenced on an Illumina HiSeq 2500 platform (Science Corporation of Gene, Guangzhou, China) under a paired-end 100 bp mode. Raw read quality was then assessed using fastp (v0.23.2) with the following parameters: a cut window size of 8 nucleotides and a qualified quality Phred score threshold of 20.

De novo CP genome assembly

The chloroplast assembly software SPAdes (v3.15.5; Bankevich et al., 2012) was used to assemble the chloroplast genome of loblolly pine, with K-mer length set to 79 and 97. The assembly’s integrity was verified by mapping reads back to the consensus sequence using Burrows-Wheeler Aligner (BWA; v0.7.17-r1188) and SAMtools (v1.14). The P. taeda cp genome was compared with other available cp genomes of Pinaceae by using the CGView Comparison Tool (CCT; Grant & Stothard, 2008). Gene annotations were performed using the Clusters of Orthologous Groups (COGs) database. The Basic Local Alignment Search Tool (BLAST) was employed to align the other genomes to P. taeda, facilitating the identification of homologous sequences. The complete annotated genome was shown as a circular map using OrganellarGenomeDRAW (OGDRAW; Greiner, Lehwark & Bock, 2019). The adenine (A) and thymine (T) distributions were measured based on the results of the AT-skew equation, as follows: AT-skew = (A−T)/(A+T).

Codon usage

Codon usage, quantified as relative synonymous codon usage (RSCU) values (Sharp & Li, 1987), was determined for all protein-coding genes across the 33 Pinaceae species. Statistical analyses of RSCU distributions were performed, and results were visualized as a heatmap using the pheatmap package (v1.0.10) in R (R Core Team, 2022).

Ka/Ks analysis

The rates of nonsynonymous (Ka) and synonymous (Ks) substitutions within the chloroplast genomes of five Pinus species were estimated using seqinR (v3.4-5; Gouy et al., 1984; R Core Team, 2022). Histograms of these substitution rates were then created using ggplot2 (v3.0.0) to visualize the distribution of Ka and Ks values.

Single nucleotide polymorphisms analysis

There were 32 P. taeda cpDNA samples, selected from the Yingde Research Institute of Forestry in Guangdong Province, P.R. China, mixed and sequenced by the Illumina Hiseq 2500 platform with a paired-end 100 bp sequencing strategy. The resulting sequences were aligned against the known reference sequence of P. taeda (GenBank accession no. NC_021440.1). Subsequently, 18 single nucleotide polymorphisms (SNPs) were selected for the detection of 33 individuals in the breeding population of P. taeda. Primer-BLAST (https://www.ncbi.nlm.nih.gov/tools/primer-blast/) was used for the primer design (Table 1), and the PCR products were sequenced using the Sanger method.

Table 1 Primers for Eighteen SNPs in Pinus taeda.

No.	Name of Primer	Location	Length (bp)	Sequence (5’-3’)	Length of Primer (bp)	GC%	PCR Annealing Temperature	SNP	
1	F1	218	540	TTCCCACAACTTTCATACCA	20	40.00%	51	ycf1_98314	
R1	757	CTTTAGGATAAGCGGGTATT	20	40.00%	
2	F2	277	423	GTCTGATTGGACCATTTGTA	20	40.00%	51	ycf2_119153	
R2	699	TCAATCTTTATGGGTCCTAC	20	40.00%	
3	F3	274	577	TGGTGGAGATGGTGAAGATG	20	50.00%	53	ycf1_101076, ycf1_101085, ycf1_101094, ycf1_101099, ycf1_101101, ycf1_101103, ycf1_101112, ycf1_101121, ycf1_101130, ycf1_101139, ycf1_101148, ycf1_101157	
R3	850	AAAGGCCATTAGACTCAGGT	20	45.00%	
4	F4	306	576	GAATATCTAAACCCTGGACT	20	40.00%	51	rpoC1_23167	
R4	881	ATTAGCTTCTCCCGAACAGA	20	45.00%	
5	F5	251	520	CTTCTCATTTCCAATCCCTG	20	45.00%	53	IGS_23593	
R5	770	GAAATGGAAGTTTGGGCTCT	20	45.00%	
6	F6	243	705	GACCCACATAAGAACAAACG	20	45.00%	53	IGS_50842, IGS_50999	
R6	947	CGGATATGTCCATGATTCACTA	22	40.91%	

Phylogenetic analysis

To resolve the phylogenetic position of P. taeda, phylogenetic analyses were conducted using both complete cp genomes and the ycf1 gene sequences from 36 samples. Sequence data were retrieved from NCBI (https://www.ncbi.nlm.nih.gov/), comprising 24 Pinus species (P. rigida, P. jaliscana, P. oocarpa, P. elliottii, P. caribaea, P. taiwanensis, P. thunbergii, P. densata, P. wallichiana, P. strobiformis, P. bungeana, P. monophylla, P. nelsonii, plus P. taeda, P. greggii, P. contorta, P. sylvestris, P. tabuliformis, P. krempfii, P. strobus, P. massoniana, P. lambertiana, P. sibirica, and P. koraiensis represented by two samples each) and the outgroup Cedrus deodara.

Both datasets were processed identically: multiple sequence alignment used MAFFT v7 (Katoh, Rozewicki & Yamada, 2019), and Maximum Likelihood trees were inferred in MEGA X (Kumar et al., 2018) with 1,000 bootstrap replicates.

Topological congruence between whole-genome and ycf1-only trees was assessed by calculating the normalized Robinson-Foulds (RF) distance using ape (v5.8.1; Paradis & Schliep, 2019) in R v4.5.1 (R Core Team, 2022).

Results

Genome comparison analysis

A total of 33,022,566 base pairs (bp) of raw reads with 38.41% GC content were generated with a paired-end 100 bp read length. The raw sequences of the P. taeda cp genome were deposited in GenBank with accession number PRJNA1159385, and the assembled sequences with accession number NC_021440.1. The size of the genome was 121,530 bp, with a GC content of 38.5% (Fig. 1). The positions of all the identified genes in the cp genome and their functional categorization are presented in Fig. 1.

Figure 1 Comparative chloroplast genome analysis of fourteen Pinaceae species against Pinus taeda.

From the outer to the inner color ring: Pinus massoniana, Pinus lambertiana, Pseudotsuga sinensis var. wilsoniana, Cedrus deodara, Abies koreana, Larix sibirica, Pseudolarix amabilis, Pinus krempfii, Keteleeria davidiana, Pinus monophylla, Tsuga chinensis, Cathaya argyrophylla, Picea sitchensis, and Nothotsuga longibracteata. Alignment method: BLASTn alignment of all genomes to Pinus taeda. Similarity scores: black (100%), red (50–99%), blue (<50%). Four outer narrow rings: Protein-coding gene positions (Pinus taeda chloroplast genome reference). Color code: Clusters of Orthologous Groups (COG). Innermost ring: AT skew of P. taeda (positive [+] = A>T; negative [–] = A.

The cp genomes of 14 species of Pinaceae (P. massoniana, P. lambertiana, Pseudotsuga sinensis var. wilsoniana, Cedrus deodara, Abies koreana, Larix sibirica, Pseudolarix amabilis, P. krempfii, Keteleeria davidiana, P. monophylla, Tsuga chinensis, Cathaya argyrophylla, Picea sitchensis, and Nothotsuga longibracteata) were selected for comparison with P. taeda by using CCT (Fig. 1). The sequence identity between P. taeda and other Pinaceae representatives was analyzed, revealing that certain regions were less conserved. For instance, ycf1 showed sequence identity lower than 70%, while ycf2 exhibited identities below 80%. Previous studies have confirmed that ycf1 has an important role in the evolution and classification of Pinus (Daniell et al., 2016; Handy et al., 2011; Georgolopoulos, Parducci & Drouzas, 2016).

Codon usage

The codon usage patterns among the 71 distinct cp protein-coding genes in P. taeda were examined (Fig. 2). The 71 protein-coding genes consisted of 20,255 codons, with the majority ending in A or T and the remainder ending in C or G. Codons terminating in A or T exhibited significantly higher RSCU values compared to those ending in C or G, as detailed in Table 1. Statistical analysis of codon usage distributions across 33 Pinaceae species revealed infrequent use of codons ending with G or C, as shown in the heatmaps (Fig. 2). Similar patterns have been reported in other cp genomes (Delannoy et al., 2011; Raubeson et al., 2007; Ying et al., 2016).

Figure 2 Relative synonymous codon usage (RSCU) heatmap of protein-coding genes across 33 species.

Color scale: RSCU values from low (blue, 0.5) to high (red, 2.0); x-x-axis: Codon patterns clustered by UPGMA (Lance & Williams, 1967; Euclidean distance).

Among the 33 analyzed Pinaceae species, codon usage patterns were largely conserved, except for P. taiwanensis, P. thunbergii, and P. koraiensis. Phylogenetic clustering revealed that P. taiwanensis and P. thunbergii formed a distinct clade, divergent from other Pinus species. Additionally, Picea and Abies occupied phylogenetically distant positions both from Pinus and from each other. The clustering results were inconsistent with traditional classification results (Gernandt et al., 2005).

Ka/Ks analysis

The study analyzed the molecular evolutionary characteristics of seven chloroplast genes (clpP, ycf1, cemA, matK, atpA, atpB, and atpI) from five species belonging to Pinus. The Ka/Ks ratios of the 10 species pairs of these genes are shown in Fig. 3. Significant differences in the Ka/Ks ratios were found between the seven chloroplast genes. Ka/Ks >1 was detected in seven species pairs for clpP and in five species pairs for matK. For ycf1 and cemA, Ka/Ks >1 was detected in three species pairs each, while the remaining species pairs had Ka/Ks ratios close to 1. For atpA, atpB, and atpI, Ka/Ks >1 was detected in only one or two species pairs, with the remaining species pairs having Ka/Ks ratios close to 1.

Figure 3 Ka/Ks ratios for the seven Chloroplast protein-coding genes among five Pinus species.

SNP analysis of the chloroplast genome

Compared with the known reference sequence, 71 SNPs and four indels were identified from the 33 individuals in the breeding population of P. taeda (Table 2). Of these, 35 SNPs were located in the gene-coding regions of 11 genes, while the remaining SNPs were found in an intergenic region. The study found that 50 SNPs had mutation frequencies below 15%, which constituted 70.42% of the total number of SNPs. The lower mutation rate indicates that the cp genome of P. taeda was conservative. Notably, three SNP loci had mutation rates between 30% and 50%, with two located in gene-coding regions (ycf1 and ycf2). Specifically, ycf1 contained 17 mutations (23.94% of the total mutation sites), and ycf2 had three SNPs, including one with the highest mutation rate of 48%. R poB had three SNPs, two of which had the same mutation rate. R poC2 had two SNPs with mutation rates at 24% and 3%. R poC1 had two SNPs with a similar mutation rate, as did psaA. The intergenic regions between matK and chlB, psbJ and petA, and rps15 and psaC contained 14 SNPs, eight of which had mutation rates higher than 10%.

Table 2 Mutation sites of the P. taeda chloroplast genome.

Coordinate	Ref	SNP/ Indels	Rate	Gene	Coordinate	Ref	SNP/ Indels	Rate	Gene	Coordinate	Ref	SNP/ Indels	Rate	Gene	
4409	A	G	0.1	matK∼chlB	39801	A	G	0.18	petA	100910	C	-GATGGTGAA	0	ycf1	
4473	C	T	0.03	matK∼chlB	41670	C	T	0.1	psaI∼accD	101076	T	G	0.01	ycf1	
4609	C	T	0.04	matK∼chlB	41678	C	G	0.06	psaI∼accD	101085	T	G	0.04	ycf1	
4910	G	T	0.09	matK∼chlB	50842	C	T	0.06	trnH-GUG∼trnT-GGU	101103	T	G	0.11	ycf1	
5514	G	A	0.27	matK∼chlB	58344	C	A	0.04	petD	101112	T	G	0.07	ycf1	
5690	G	A	0.12	matK∼chlB	64071	T	C	0.04	rpl22	101121	T	G	0.1	ycf1	
9984	A	C	0.23	trnG-GCC	65304	C	A	0.15	rpl2	101130	T	G	0.06	ycf1	
10349	A	C	0.06	trnG-GCC	70068	T	G	0.04	trnT-UGU∼rps4	101139	G	T	0.4	ycf1	
11437	G	A	0.1	atpA	71282	A	C	0.04	trnS-GGA∼ycf3	101157	T	G	0.09	ycf1	
19229	G	A	0.24	rpoC2	71422	C	G	0.12	trnS-GGA∼ycf3	101166	T	G	0.02	ycf1	
20388	C	T	0.03	rpoC2	74477	T	G	0.23	ycf 3∼psaA	102292	A	C	0.36	ycf1∼rps15	
22162	A	T	0.15	rpoC1	74535	A	C	0.05	ycf 3∼psaA	102658	C	T	0.06	rps15∼psaC	
23394	T	G	0.1	rpoC1	75726	C	T	0.04	psaA	103514	C	T	0.06	rps15∼psaC	
24094	C	T	0.04	rpoB	76044	A	G	0.04	psaA	103620	G	T	0.18	rps15∼psaC	
24666	G	A	0.04	rpoB	81006	T	G	0.04	trnfM-CAU∼psbZ	103785	G	A	0.07	rps15∼psaC	
26193	C	A	0.12	rpoB	83527	G	T	0.09	psbD∼trnT-GGU	104282	G	A	0.1	rps15∼psaC	
29448	T	G	0.11	psbM∼trnD-GCA	84756	A	G	0.05	tRNA-Thr∼rrn16	104944	G	A	0.2	psaC∼ccsA	
29449	T	G	0.09	psbM∼trnD-GCA	97585	T	G	0.08	ycf1	105020	T	C	0.13	psaC∼ccsA	
30550	G	T	0.07	trnE-UUC∼clpP	98169	T	A	0.05	ycf1	105446	A	C/G	0.11/0.01	psaC∼ccsA	
34091	T	G	0.1	psaJ∼trnP-UGG	100294	A	C	0.26	ycf1	109795	A	C	0.28	rpl 32∼trnV-GAC	
37238	T	A	0.24	psbJ∼petA	100387	T	G	0.35	ycf1	113045	T	G	0.22	rps 7∼trnL-CAA	
37239	T	A	0.24	psbJ∼petA	100523	G	T	0.02	ycf1	114839	T	C	0.09	ycf 2	
37240	T	A	0.24	psbJ∼petA	100598	A	C	0.13	ycf1	114997	A	T	0.04	ycf 2	
37263	A	-ATCT	0	psbJ∼petA	100730	A	G	0.02	ycf1	117038	T	+TCTTCC	0	ycf 2	
38094	C	-GAAG	0	psbJ∼petA	100732	G	T	0.02	ycf1	119153	G	T	0.48	ycf 2	

Individual distinction of the breeding population

A total of 18 SNPs were selected to detect 33 individuals from the breeding population of P. taeda. Of these, 13 SNPs were mapped to ycf1. Analysis revealed that 32 individuals exhibited SNPs across at 15 different loci, and 23 individuals could be distinguished based on these SNPs. Only one individual’s genome matched the reference genome, as shown in Table 3. Among the 23 distinguished individuals, one additional SNP was identified, and seven SNPs were found to be the most prevalent.

Table 3 Fifteen single nucleotide polymorphisms (SNPs) for individual identification in the Pinus taeda breeding population.

Gene	rpoC1	IGS	IGS	IGS	ycf1	ycf1	ycf1	ycf1	ycf1	ycf1	ycf1	ycf1	ycf1	ycf1	ycf2	
Pos	23167	23593	50842	50999	98314	101076	101085	101103	101112	101121	101130	101139	101148	101157	119153	
Ref	G	C	C	G	G	T	T	T	T	T	T	G	G	T	G	
013	–	–	–	–	T	–	–	–	–	–	G	–	T	–	T	
023	–	A	T	A	–	–	–	G	–	–	–	T	–	–	–	
024	A	A	T	A	–	–	–	–	–	–	–	T	–	–	–	
026	–	A	T	–	–	–	–	G	–	–	–	T	–	–	–	
222aA	A	A	T	–	–	–	–	–	–	G	–	–	T	–	–	
243	–	–	–	–	–	–	–	G	–	–	–	–	–	–	T	
248	–	–	–	–	–	–	–	–	–	–	–	–	T	–	–	
252	–	–	–	–	T	–	–	–	–	–	–	–	–	–	T	
257	–	–	–	–	N	–	–	–	–	–	–	T	–	–	T	
259	–	A	T	A	T	–	–	G	–	–	–	T	–	–	–	
262	–	–	–	–	T	–	–	–	–	–	–	T	–	G	T	
N3	–	–	–	–	–	–	–	–	–	–	–	T	–	–	–	
G01	–	N	T	A	–	–	–	G	–	–	–	T	–	–	–	
G10	–	–	–	–	–	–	–	G	–	–	–	–	–	–	T	
G9	–	–	–	–	–	–	–	G	–	–	–	–	–	–	T	
P012	–	–	–	–	T	G	G	G	G	–	G	–	–	–	T	
P034	–	–	–	–	–	–	–	–	–	G	–	T	–	–	–	
P043	–	–	–	–	–	–	–	–	–	–	–	–	–	G	T	
P052	A	A	T	A	–	–	–	–	–	G	–	T	–	–	–	
P054	–	–	–	–	–	–	–	–	–	–	–	–	–	G	T	
P063	–	–	–	–	–	–	–	G	–	–	G	–	–	–	–	
P064	–	–	–	–	–	–	–	–	–	–	–	–	–	G	T	
Q16	–	–	–	–	–	–	–	–	–	G	–	–	–	–	T	
Q26	–	–	–	–	–	–	–	–	–	–	G	–	–	–	T	
Q6	–	–	–	–	T	–	–	–	–	–	–	–	–	–	T	
S11	A	A	T	A	–	–	–	G	–	G	–	T	–	–	–	
S3	–	–	–	–	–	–	–	–	–	–	G	–	–	–	–	
S6	–	–	–	–	–	–	–	–	–	–	N	–	–	–	–	
W05	–	–	–	–	T	–	–	–	–	–	–	T	T	–	T	
W11	–	–	–	–	–	–	–	–	–	–	–	T	–	–	–	
W13	–	–	–	–	–	–	–	–	–	–	–	–	–	–	–	
W16	A	A	T	A	–	–	–	–	–	–	–	–	–	–	–	
W28	–	–	–	–	–	–	–	–	–	–	–	–	–	–	T	

Furthermore, three additional SNPs were found in 14 individuals, accounting for 60.87% of the distinguishable SNPs. A total of 13 individuals had two SNPs, which represented 39.40% of the total number of individuals (33). Compared to previous studies, the cpDNA markers had a comparatively high level of variability (Walter & Epperson, 2010). The chloroplast simple sequence repeats (cpSSRs) of P. slvestris revealed a high level of intra-populational polymorphism across the entire range (Semerikov et al., 2014). Additionally, the variation among populations accounted for 99% of the total variance (Pazouki et al., 2016).

Additionally, the remaining 10 individuals that could not be distinguished, each of which had only one or two SNPs, could be classified into four groups. Introducing more SNPs could potentially help distinguish these individuals. Specifically, individuals P043, P054, and P064 exhibited identical SNP profiles. Given their common origin, they may descend from the same paternal lineage. Similarly, individuals G09, G10, and 243 also had the same SNPs. G09 and G10 had the same origin, while 243 was selected from a different stand. The loblolly pine in China, which were all introduced from the southern United States, might be related to each other.

Phylogenetic insights from cp Genomes and ycf1

Phylogenetic reconstruction using complete cp genomes resolved the 24 Pinus species and the outgroup C. deodara into two well-supported clades (Clade A and Clade B; Fig. 4A). The topology contained 33 nodes, with 26 nodes achieving maximal bootstrap support (100%), while the remaining nodes attaining values ranging from 64% to 99%. This bifurcation is consistent with the canonical classification proposed by Gernandt et al. (2005). Notably within Clade A, P. taeda and P. rigida converge into the same branch. P. taeda emerged as a focal lineage closely affiliated with P. jaliscana, P. oocarpa, P. greggii, P. elliottii, and P. caribaea, corroborating clade relationships reported by Duan et al. (2016) and Zeb et al. (2019); Zeb et al. (2022).

Figure 4 Phylogenetic trees of Pinaceae from 36 plastomes and ycf1.

Phylogenetic relationships inferred by maximum likelihood (ML) from 36 complete chloroplast genome sequences representing 25 species of Pinaceae. (A) Whole-chloroplast genome phylogeny. (B) ycf1 gene phylogeny. Major clades are indicated: Clade A, subgenus Pinus (two-needle pines); Clade B, subgenus Strobus (single-needle pines). The red dot represents the position of P. taeda (NC_021440.1).

Analysis of ycf1 gene sequences from the same chloroplast genomic resources recovered an overall congruent phylogenetic architecture (Fig. 4B), despite minor topological variations. The resultant tree likewise recovered Clades A and B across 33 nodes, with 16 nodes exhibiting 100% bootstrap support and the remaining nodes having bootstrap values ranging from 52% to 99%. While P. jaliscana, P. oocarpa, and P. greggii retained their close clustering with P. taeda within Clade A, P. elliottii and P. caribaea failed to group with this lineage. Instead, they occupied distinct positions with bootstrap supports of 92% at their respective nodes.

Quantitative assessment of topological concordance revealed minimal divergence between the whole-plastome and ycf1-based phylogenies, evidenced by a normalized RF distance of 0.15. This strong congruence highlights the efficacy of ycf1 as a single genomic marker for resolving genus-level evolutionary relationships in Pinus.

Discussion

The complete cp sequence of P. taeda, a dominant non-native plantation species in China, has been assembled, annotated, and analyzed. This species is a valuable resource for investigating the intra- and interspecific evolutionary history of plants (Birky Jr, 1978; Birky Jr, 2001; Chase et al., 1993; McCauley, 1995; Newton et al., 1999; Provan, Powell & Hollingsworth, 2001; Petit et al., 2003). The cp genome consisted of 121,530 bp and lacked inverted repeats, which are commonly found in most angiosperms.

Comparative genomic analysis between P. taeda and 14 species of Pinaceae showed that certain regions, such as ycf1, were less conserved. This finding was confirmed in a previous study (Jiang et al., 2017; Li et al., 2020; Chen et al., 2020; Zhang et al., 2023). The highest sequence diversity was identified in two regions of ycf1. These regions can be targeted in all pine subsections using three primer combinations (Parks, Liston & Cronn, 2011; Handy et al., 2011). Recently, ycf1 has been identified as an essential component of the protein translocon at the chloroplast’s inner envelope membrane (Kikuchi et al., 2013). Seven variable sites and nine haplotypes were found in an 840 bp fragment of the DNA-coding region of ycf1. These variable sites and haplotypes have been evaluated in the Pinus subsection Australes and have unquestionable value for studying evolution in the group (Ortiz-Martínez & Gernandt, 2016). The present study corroborates that ycf1, with 17 mutations identified, plays a significant role in individual distinction in P. taeda.

It is probable that the chloroplast genes of P. taeda exhibit large differences in the natural selection pressure during the evolutionary process. This is in comparison to the chloroplast genes of other species belonging to the same genus (Kober & Pogson, 2013; Nasrullah et al., 2015; Kwon et al., 2016). However, most species of the same genus have similar codon biases and are basically clustered together. These findings suggest that the codon usage bias of chloroplast genes is closely related to the genetic similarity among species, reflects the evolutionary relationships between species, and could provide additional insights when used in conjunction with phylogenetic analysis to investigate the evolutionary relationships and molecular evolutionary mechanisms of species.

The current study surveyed the evolutionary characteristics of seven chloroplast genes across five species of Pinus. The findings indicated that for clpP and matK, most species pairs exhibited Ka/Ks ratios >1. In contrast, ycf1 and cemA showed ratios approximately half of that value, while the remaining species pairs for these genes had ratios near 1. For atpA, atpB, and atpI, Ka/Ks >1 was observed in only one or two species pairs, while the ratios for the majority of pairs were well below 1. These results suggest that clpP and matK experienced predominantly positive selection, ycf1 and cemA experienced either positive or neutral selection, and atpA, atpB, and atpI were under purifying selection during Pinus evolution (Yang, 1998; Zhang, 2005). This variation may be attributed to the encoded proteins playing different roles in growth and metabolism processes (Hahn, 2005; Ransay, Rieseberg & Ritland, 2009; Cork & Purugganan, 2004). cpSNP markers helped distinguish individuals among populations of loblolly pine. Distinguishing individuals is crucial for identifying the male parent of open- or mixed-pollination offspring. Given the paternal inheritance of cpDNA in Pinus, the male parent can be identified by using the molecular marker developed from SNPs in the cp genome. Fifteen SNPs successfully distinguished 23 individuals within the P. taeda breeding population. The inability to distinguish the remaining individuals suggests they may be closely related. P. taeda was introduced to China nearly 120 years ago, and the individuals in the breeding population of the current study were selected from plantations or had just been introduced from the southern United States. The genetic relationships among individuals in the P. taeda breeding population were previously undefined. The study of the SNPs in the cp genome may provide a new method for paternal identification in the breeding program of P. taeda.

Chloroplast genomes are valuable tools for plant phylogenetic analysis, with important insights often coming from the examination of protein-coding genes (Eckert & Hall, 2006; Gernandt, Liston & Piñero, 2003; Moore et al., 2010; Parks, Cronn & Liston, 2012; Zhu et al., 2016a; Zhu et al., 2016b). These earlier studies provided the foundation for the current investigation into the loblolly pine chloroplast genome. This study performed a phylogenetic analysis using the maximum likelihood (ML) method. The dataset comprised 36 chloroplast genomes and the ycf1 gene from 24 Pinus species, with C. deodara as the outgroup. Phylogenetic trees based on both the complete chloroplast genomes and the single ycf1 gene consistently resolved the 25 species (which were represented by 36 samples) into two major clades: the subgenus Strobus (single needle section, clade B) and subgenus Pinus (double-needle section, clade A). Particularly significant is the placement of P. taeda within Clade A, where it forms a sister relationship with P. rigida while maintaining close affinity to P. jaliscana, P. oocarpa, and P. greggii. This configuration aligns with and extends earlier findings by Zeb et al. (2019); Zeb et al. (2022), resolving previous ambiguities regarding loblolly pine’s phylogenetic position.

Notably, analysis of the ycf1 gene region alone recovered a highly congruent phylogenetic architecture (Fig. 4B), despite minor topological variations affecting P. elliottii, P. caribaea and C. deodara placements. The high degree of concordance between whole-plastome and single-gene topologies-quantified by a normalized RF distance of 0.15-validates ycf1 as a phylogenetically informative marker for genus-level reconstructions in conifers. This finding substantiates earlier suggestions (Zhu et al., 2016a; Zhu et al., 2016b) regarding the utility of protein-coding genes, while demonstrating that targeted sequencing of ycf1 has comparable resolution to resource-intensive complete plastome analyses in resolving major clades (e.g., subgenera) in Pinus.

Conclusions

We sequenced and deposited the 121,530-bp chloroplast genome of Pinus taeda (GenBank: NC_021440.1). Regions such as ycf1 and ycf2 showed markedly lower identity (<70% and <80%, respectively) than 14 other Pinus species. Phylogenomic analyses placed P. taeda within the diploxylon clade, sister to P. rigida. Comparative codon-usage patterns across 33 Pinaceae species revealed signatures of natural selection. In the breeding population, 71 cpSNPs distinguished 72% of individuals; the single-copy gene ycf1 recapitulated whole-plastome relationships (normalized RF = 0.15), validating it as an efficient barcode. These cpSNPs and the mutation map provide readily applicable markers for paternal identification in P. taeda breeding programs, while the hypervariable ycf1 gene serves as a highly informative marker for future genus-wide phylogenetic and germplasm studies.

Additional Information and Declarations

Competing Interests

Author Contributions

Data Availability

The authors declare there are no competing interests.

Ling Wang conceived and designed the experiments, performed the experiments, analyzed the data, prepared figures and/or tables, authored or reviewed drafts of the article, and approved the final draft.

Jipeng Mao analyzed the data, prepared figures and/or tables, and approved the final draft.

Kaibin Jiang analyzed the data, prepared figures and/or tables, and approved the final draft.

Zhengyu Wu analyzed the data, prepared figures and/or tables, and approved the final draft.

Chunxin Liu analyzed the data, prepared figures and/or tables, and approved the final draft.

Huagong Ning performed the experiments, prepared figures and/or tables, authored or reviewed drafts of the article, and approved the final draft.

Haibiao Zhou analyzed the data, authored or reviewed drafts of the article, and approved the final draft.

Jiehu Chen analyzed the data, authored or reviewed drafts of the article, and approved the final draft.

Shaowei Huang analyzed the data, prepared figures and/or tables, and approved the final draft.

Tianyi Liu conceived and designed the experiments, analyzed the data, prepared figures and/or tables, authored or reviewed drafts of the article, and approved the final draft.

The following information was supplied regarding data availability:

The raw data is available at CNGBdb: CNP0006272.

The raw data is available at GenBank: PRJNA1159385 and NC_021440.1.

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
