# Peer review of "cpSNP discovery and genotyping for a Pinus taeda breeding population with targeted comparison to related conifers"

_PeerJ, doi:10.7717/peerj.20092_

## Round 0.1 · original submission · Major Revisions

Dear Authors

Your paper has been reviewed by two experts and we received mixed reviews from them. However I would like to give you another chance to revise the paper following the reviewers suggestions. If you disgree with any suggestion give us a valid justification. Also please provide pont-by-point response to the comments. As suggested by reviewer 1 it is mandatory to deposit the original data and provide the relevant information and link that enables anyone should be able to access the data.

Reviewer 1 ·

Basic reporting

Writing Quality: The manuscript needs significant revision for writing quality. The sentences are highly fragmented and not well linked. For example, the abstract lacks cohesion, with sentences not connecting logically to each other. The aim of the study is not clearly defined, and the experiments are explained without context.
Figure Quality: The figures, especially Figure 1 and Figure 2, do not have sufficient quality.
Insufficient Introduction and Background: The article does not include a sufficient introduction and background to demonstrate how the work fits into the broader field of knowledge. The introduction is very short and lacks a sufficient literature review.
Structural Problems: The manuscript has structural problems, including incorrect spacing and missing punctuation at the end of sentences.

Experimental design

1. The authors used Illumina technology to sequence the chloroplast genome instead of long-read technology such as PacBio. The rationale for this choice is not provided, despite long-read technologies being more suitable for assembling complete chloroplast genomes due to their ability to generate longer reads, which can span repetitive regions and improve assembly accuracy.
2. How many reads do you find in total at various coverage depths for full genome sequencing? How do these numbers compare with the number of randomly selected reads used in the assemblies presented here.
3. The manuscript does not specify how the read quality was assessed or whether the reads were trimmed. If trimming was performed, details on the trimming process, including the software used and the parameters set, are not provided.
4. The manuscript does not include a phylogenetic analysis, which is a critical component for understanding the evolutionary relationships between the chloroplast genomes of P. taeda and related species.

Validity of the findings

1. The manuscript does not specify where the original data has been deposited. For transparency and reproducibility, the authors should provide information on the repository where the sequencing data and other relevant datasets have been deposited, along with accession numbers or links to access the data.
2. the discussion section lacks depth and coherence, requiring improvement to provide a more comprehensive interpretation of the findings and their implications The discussion should covers all the key points - structure, characteristics, comparisons etc.

·

Basic reporting

Wang et al. Sequenced assembled and analyzed the cp genome of Pinus taeda, and was compared with the available cp genome of Pinidae. This study is of interest to the scientific community.
On the other hand, please see below some comments that may be of help:
• Line 25. Delete “in”
• Line 26. Space after “P.” Check the entire manuscript.
• Line 28. Change “were” to “was.
• Check the manuscript for English errors (grammar)
• Line 32. There is no need to include GenBank No.
• Introduction. This section needs improvement. The authors should describe the importance of this species in China, then mention gaps on the study of this species, importance to conduct their study and so on.
• DNA Extraction & Sequencing. Was the DNA verified (quality and quantiy)? If so, how did the authors did it?
• DNA Extraction & Sequencing. Do samples have a voucher code?
• DNA Extraction & Sequencing. How did you isolate cpDNA?
• DNA Extraction & Sequencing. Where were samples sequenced?
• Line. 86. Add citation of the program. Check the entire manuscript.
• Line 86. Did the authors use default parameter? More details are needed.
• Line 89. How did you obtain the circular genome? Which software was used to get the graphic?
• Line 100. Authors have to indicate R program was used. Add also citation.
• Line 100, 101. There is no need to add weblinks.
• Line 103, 106. Do not begin a sentence with a number. Check the entire manuscript.
• Figures 1, 2, 3. Improve the resolution of the figure. Scientific name of plant should be in italics.
• Sequence data and aligned dataset. Please indicate weblink.

Experimental design

More details are needed in the Materials and Methods section.

Validity of the findings

It is OK

---

## Round 0.2 · Minor Revisions

The revised version of the manuscript has been reviewed by me and reviewer 2. The authors have done a great job in revising the paper, and its quality has improved significantly.

However, one of the Section Editors feels that the manuscript still needs to be edited for English. Please perform another edit of the whole document, ensuring the language and grammar is improved.

·

Basic reporting

I am satisfied with the author’s responses to my questions/issues raised in my initial review. I recommend that the revised paper be accepted.

Experimental design

No comment

Validity of the findings

No comment

Additional comments

No comment

---

## Round 0.3 · Major Revisions

As noted by one of her Section Editors, the quality of the English language in the manuscript needs *significant* improvement before this article can progress. You must have it comprehensively edited or it may be rejected.

Some specific observations from the Section Editor are as follows, but this is NOT a comprehensive list, they just commented on the first section of the manuscript:

"Firstly, the title does not make sense: "Comparative analysis of chloroplast genome with related species and SNP analysis among breeding population of Pinus taeda". With related species to what? Better title, "SNP analysis among breeding population of Pinus taeda and comparative analysis of its chloroplast genome with related species."

The abstract, 1st sentence: "Loblolly pine is the most important commercial tree species in the southern United States and a significant non-native plantation species in China with the genetic improvement program for P. taeda has been carried out in South China for 30 years in." This is not a complete sentence and does not make sense.

Here is what I think that the authors' meant: "Loblolly pine is the most important commercial tree species in the southern United States and a significant non-native plantation species in China. The genetic improvement program for P. taeda has been carried out in South China for 30 years."

To continue, I recommend to move specifics of what was done to the methods and revise (here are some suggested edits): "In this study, we sequenced, assembled, and analyzed the chloroplast genome of P. taeda and compared it to 14 other chloroplast genomes of Pinidae spp. While some regions were conserved, others, such as ycf1, had less than 70% identity. Codon usage among 33 species of Pinidae was compared in 71 distinct chloroplast protein coding genes and 20,255 codons. Heatmaps of 33 species indicated that codons ending with G or C were infrequent. All Pinus species exhibited similar codon usage patterns except for P. staiwanensis, P. thunbergia, and P. koraiensis, and it is likely that the chloroplast genes of these species were subject to large differences in evolutionary natural selection pressure. The rates of nonsynonymous (Ka) and synonymous (Ks) substitutions in the chloroplast genomes among five species of Pinidae were estimated, and single nucleotide polymorphisms (SNPs) were selected to genotype 33 individuals from the P. taeda breeding population. Compared with the reference sequence, 71 SNPs were identified in the chloroplast genome of P. taeda, with mutation frequencies below 15% in 50 SNPs, and 30-50% mutation frequencies in only four SNPs. In addition, 17 mutations were found in ycf1, accounting for 24% of the total mutation sites. A total of 32 individuals from the breeding population of loblolly pine had 1-7 SNPs in 15 loci that were distinguishable in 23, and only one individual aligned exactly to the reference genome. Phylogenetic analysis of the chloroplast genome sequences of P. taeda and other 15 species of Pinidae spp. revealed that P. taeda is more closely related to P. elliottii. The complete sequences and the mutation map of the chloroplast genomes of P. taeda enhance our understanding of the characteristics of Pinus chloroplast genomes and demonstrate the potential utility of chloroplast SNPs in paternal identification and genetic relationship determination within the breeding program of Pinus taeda."

The entire manuscript still needs significant editing.

---

## Round 0.4 · Major Revisions

The manuscript requires major revisions. Firstly, authors need to improve readability—many grammar errors and unclear sentences (especially in abstract/introduction) affect clarity. Please, consider professional editing.
It is also important to better highlight novelty. Previous works already reported P. taeda chloroplast genome. Explain clearly how your work adds new data (e.g., SNPs), methods, or comparisons.

It is important to fix the title-content mismatch. The title suggests a broad comparative study, but results focus only on P. taeda SNPs. Either expand analyses (e.g., IR boundaries, GC content across species) or adjust the title to reflect the actual focus.

Phylogenetics needs to be deeply revisited: authors need to adopt Maximum Likelihood/Bayesian methods instead of Neighbor-Joining for genome-scale data. Include hypervariable regions (e.g., *ycf1/ycf2*) and more public genomes per species.

Lastly, it is necessary to improve figure/table captions. Ensure references follow journal formatting.

**Language Note:** We note that you have already obtained editing services from PeerJ. However, PeerJ copyeditors are not subject specialists and only edit the language. It is your responsibility to ensure the text conveys your intended message. It might be helpful to enlist the help of a colleague with relevant subject knowledge.

·

Basic reporting

Professional article structure, figures, and tables.

The descriptions of the tables require improvement. For instance, the caption for Table 3 should begin with a descriptive sentence rather than a numeral.

Experimental design

Introduction
Lines 81–88 should more clearly articulate the novelty of this study. Given the existence of similar prior work, such as the chloroplast genome study of Pinus taeda published in PLOS ONE (https://journals.plos.org/plosone/article?id=10.1371/journal.pone.0192966), the authors are encouraged to highlight the specific contribution and originality of their research. This includes clarifying how their approach, taxonomic sampling, or genomic analysis differentiates this study from existing literature.

Methods are described with sufficient detail & information to replicate

Materials and Methods
The phylogenetic analyses appear to be limited to basic methods. The use of the Neighbor-Joining (NJ) algorithm (Figure 4) is not sufficiently robust for evolutionary inference at the genomic scale. The authors are strongly encouraged to consider more rigorous methods such as Maximum Likelihood (ML), Maximum Parsimony (MP), or Bayesian Inference (BI) for phylogenetic reconstruction. These approaches would better support the analysis of both interspecific and intraspecific evolutionary relationships. Additionally, incorporating more than one chloroplast genome sequence per species (if available from public databases such as NCBI) would enhance the resolution and reliability of the phylogenomic framework.

Validity of the findings

Impact and novelty not assessed.

Results
Furthermore, it would be valuable to include a phylogenetic tree based on hypervariable regions such as ycf1 or ycf2, which have shown high discriminatory power among Pinus species and could serve as potential molecular markers for species delimitation or identifying reproductively isolated populations.

Additional comments

The references and in-text citations should be carefully revised to conform to the journal’s formatting requirements.
Moreover, the manuscript currently focuses more on SNP analysis than on comprehensive genome comparison, despite the latter being implied in the title. To fully meet the stated objectives, the authors should include additional analyses such as the comparison of repetitive elements, IR boundary variation, genome size, gene content, GC content across coding and non-coding regions, and a more robust phylogenomic analysis using ML or BI methods.

---

## Round 0.5 · Minor Revisions

The revised manuscript shows substantial effort to address reviewers, and I agree with most of the rebuttal. Many concerns have been addressed, and the manuscript is stronger in terms of analysis and structure.

However, some writing issues remain and affect clarity and precision. In the Abstract, the sentence “This study provides a validation of the single-copy gene ycf1 as a robust and low-cost and effective phylogenetic marker for conifer genus-level reconstruction” is wordy and redundant, it might be more concise. In the Introduction, the sentence beginning “Conifers are of immense ecological and economic value…” is overloaded with clauses, while “a growing body of evidence confirms that ycf1 is a hypervariable locus” is imprecise for a scientific article. In Methods, long constructions such as the description of Illumina sequencing and fastp parameters reduce readability, and terms like “half-synonymous codons” in the Codon Usage section are nuclear and could be splitted in shorter sentences. In Results, sentences such as “the remaining 10 individuals that could not be distinguished… could be classified into four groups” are structurally awkward, and vague statements like “The genetic relationship… was not clear” should be refined. In the Conclusions, the expression “ultra-efficient, single-locus barcode” could be less promotional. Finally, the use of “preliminary” in the title is not suitable for a full-length article, as it suggests the work may not yet be ready for publication. A final careful revision focusing on clarity, conciseness, and scientific neutrality is still required.

·

Basic reporting

No comment

Experimental design

No comment

Validity of the findings

No comment

Additional comments

The authors have satisfactorily addressed all the comments raised. The manuscript is now complete and demonstrates improved clarity and scientific rigor.

---

## Round 0.6 · accepted · Accept

The reviewers’ comments have been adequately addressed in the revised version. As the original reviewers were not invited for this round, the assessment has been conducted by the editor. The manuscript is considered suitable for publication.